# PreWoMe: Exploiting Presuppositions as Working Memory for Long Form Question Answering

**Wookje Han**
Columbia University
wookje.han@columbia.edu

**Jinsol Park**
Carnegie Mellon University
jinsolp@cs.cmu.edu

**Kyungjae Lee**[*]
LG AI Research
kyungjae.lee@lgresearch.ai

## Abstract

Information-seeking questions in long-form question answering (LFQA) often prove *misleading* due to *ambiguity* or *false presupposition* in the question. While many existing approaches handle misleading questions, they are tailored to limited questions, which are insufficient in a real-world setting with unpredictable input characteristics. In this work, we propose PreWoMe, a unified approach capable of handling any type of information-seeking question. The key idea of PreWoMe involves extracting presuppositions in the question and exploiting them as working memory to generate feedback and action about the question. Our experiment shows that PreWoMe is effective not only in tackling misleading questions but also in handling normal ones, thereby demonstrating the effectiveness of leveraging presuppositions, feedback, and action for real-world QA settings.

## 1 Introduction

Answering information-seeking long-form questions has recently shown significant progress (Fan et al., 2019; Krishna et al., 2021; Xu et al., 2023). However, users' questions in the real-world may often *mislead* language models (LMs) to output misinformation. For example, users ask questions with **false presuppositions** (FP) (Kim et al., 2022; Yu et al., 2022) which can induce hallucinations if LMs believe the presuppositions are true (Maynez et al., 2020). Also, users often ask questions that do not have a single and clear answer, *i.e.*, an **ambiguous** question, which is difficult for LMs to identify (Min et al., 2020; Stelmakh et al., 2022).

Figure 1 shows examples of two types of misleading questions: (a) an ambiguous question that requires considering prior knowledge ("two different wars between Italy and Ethiopia") to answer properly, and (b) a question with FP ("somebody

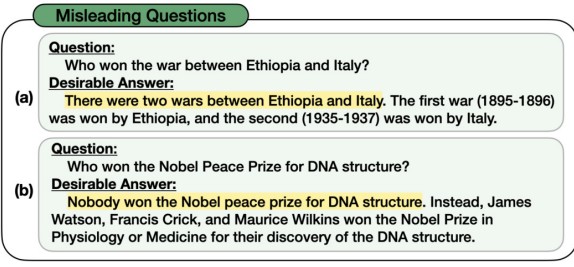

Figure 1: Two types of misleading questions: (a) having ambiguity, and (b) false presupposition. Highlights show essential cues to avoid misleading.

won the Nobel peace prize for DNA structure"), which should be corrected before answering the question (Yu et al., 2022; Kim et al., 2022).

Desirable answers to misleading questions require discerning and resolving misleadings. A previous work by Amplayo et al. (2022) focuses on ambiguous questions, by introducing query refinement prompts that encourage LMs to consider multiple facets of the question. Another work (Kim et al., 2021) tackles questions containing FP, by extracting and verifying presuppositions. While such works assume only one specific type of misleading factor, in a real scenario, the users ask various types of questions, which can be misleading or normal (*i.e.*, non-misleading).[1] This makes prior single-type-tailored approaches hard to be deployed in real-world QA settings.

In this paper, we emphasize the role of presuppositions in handling *any type* of information-seeking questions in a *unified* way. Presuppositions are crucial building blocks of questions and play an essential role in understanding the meaning of the question (Ge, 2011; Duží and Číhalová, 2015). This suggests that exploiting presuppositions is promising in handling misleading questions, while generally applicable to any type of question. Based on this, we propose a new approach, **PreWoMe**

---

[*]Corresponding author

[1]Throughout this paper, we use the term *misleading questions* for questions that are ambiguous or have FP.

(**Pre**supposition as **Wo**rking **Me**mory), that handles *any type* of information-seeking question in a *unified* way without any parameter updates.

The main idea of PreWoMe is to extract presuppositions and use them as working memory to generate analysis and directions for answering the question. Our contributions can be summarized as follows:

- We analyze the performance of recent large language models (LLMs), GPT-4 and GPT-3.5, on misleading questions. To the best of our knowledge, we are the first to explore the performance of LLMs this large on misleading questions.
- We propose PreWoMe, a new approach designed for real-world QA settings that 1) is capable of handling any type of information-seeking question, and 2) does not entail any parameter updates, making it easily adaptable to LLMs.
- We propose using presuppositions as a working memory when answering information-seeking questions.

## 2 Background

**Presupposition** Presuppositions are conditions that are believed to be true by speakers in a discourse (Yu et al., 2022). Thus, in order for an utterance to be *appropriate*, the presuppositions should be true. For example, if someone said *I care for my sister*, then it can be assumed that the speaker has a sister. If not, the utterance would be *inappropriate*. In this paper, we suggest using presuppositions to solve an information-seeking QA task.

**Chain-of-thought Prompting (CoT)** Wei et al. (2022) explores how thinking with intermediate steps greatly improves LM's performance. Chain-of-thought Prompting effectively enhances the reasoning abilities of LMs, particularly in mathematical and logical reasoning tasks. This technique involves guiding the model through a series of interconnected thought processes. Inspired by this concept, we propose a novel approach that leverages presupposition, feedback, and action, serving as intermediate steps, which boosts the robustness of LMs in handling open-ended questions.

**Working Memory** Working memory (Miller et al., 2017) is the immediate information buffer that is accessed while performing conscious tasks (Li et al., 2022). We make presuppositions function as working memory by extracting and feeding them into the model along with the question as intermediate steps.

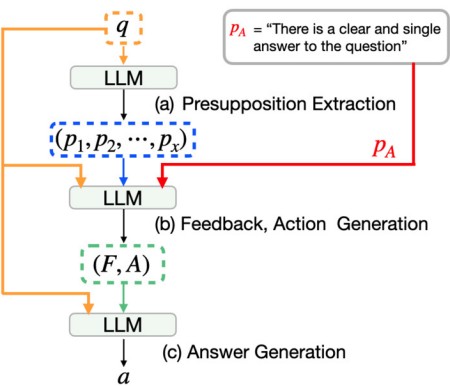

Figure 2: Overall pipeline of PreWoMe. $q$ is the given question, $(p_1, p_2, ..., p_x)$ are the model-generated presuppositions. $(F, A)$ is feedback and action.

**Self-correcting Approaches** While LLMs occasionally struggle to generate accurate answers without hallucination, recent works suggest that LLMs can enhance accuracy by *self-correcting* their responses through iterative prompting (Madaan et al., 2023; Shinn et al., 2023; Saunders et al., 2022; Press et al., 2022; Weng et al., 2023). Such approaches are based on the idea that verifying correctness is easier than directly generating an accurate answer. Inspired by this idea, we hypothesize that 1) LLMs are capable of generating and verifying the generated presuppositions implicit in a given question, and 2) guiding LLMs to generate answers *after* the verification will yield better results than generating answers directly.

## 3 PreWoMe

We introduce PreWoMe, a new approach that answers any type of long-form information-seeking question in a unified way. PreWoMe follows three intermediate steps, as illustrated in Figure 2.

### 3.1 Presupposition Extraction

To exploit presuppositions as working memory, PreWoMe first generates presuppositions contained in the question $q$. Motivated by the few-shot learning capability of LLMs (Brown et al., 2020), we feed $q$ with $N$ (question, presupposition) pairs as few-shot examples within a designed prompt (See Appendix A). The LLM generates multiple presuppositions $(p_1, p_2, \cdots, p_x)$ in $q$ using this input (Figure 2(a)).

In addition, we append a manual presupposition $p_A$: *"There is a clear and single answer to the question"*, as shown in Figure 2, which is a common presupposition of all normal questions. If $p_A$ is recognized as False in the next step, the question

| Type | (a) Ambiguous | (b) False Presupposition |
|---|---|---|
| Question | Who won the war between Ethiopia and Italy? | Who is the only Indian to win the Oscar for music? |
| Presuppositions | 1) There was a war between ethiopia and italy.
2) Some country won the war.
A) There is a clear and single answer to the question. | 1) There is a category for music in the Oscars.
2) There is only one Indian who won the Oscar for music.
A) There is a clear and single answer to the question. |
| Feedback | There were two wars between Italy and Ethiopia, with different winners. Therefore, the question **contains a false presupposition that there is a clear and single answer to the question.** | The question **contains a false presupposition that there is only one Indian who won the Oscar for music.** |
| Action | Your answer should include the winner of the war between Ethiopia and Italy for **each war respectively in detail**. | **Correct the false presupposition** that there is only one Oscar winner Indian and respond based on the corrected presupposition. |

Table 1: Examples of misleading questions and their intermediate steps. Boldface in feedback and action are parts that distinguish different types of questions.

is classified as *misleading* due to its ambiguity.

## 3.2 Feedback and Action Generation

Motivated by Yao et al. (2022) that explores the effectiveness of reasoning processes and task-specific actions, we make LLMs to generate *feedback* and *action* for $q$ (Figure 2(b)) by exploiting extracted presuppositions as working memory. We concatenate $q$ with extracted presuppositions and $p_A$, and feed it as part of an input to the LLM. We use $N$ (question, presupposition, feedback-action) triplets as few-shot examples within a designed prompt (See Appendix A).

Table 1 shows examples of the generated feedback and action for two types of misleading questions. The *feedback* is the description of whether the given input contains FP or not. It can be seen that both types of misleading questions can be handled in a unified way – by indicating which presupposition is false. If the LLM determines that $p_A$ is false, the feedback provides the hint that question $q$ is ambiguous. If other presuppositions are determined to be false, then the corresponding feedback states that question $q$ contains FP. For normal questions, the feedback informs that all presuppositions are valid (See Appendix B). The feedbacks can be viewed as a classifier that determines the type of input question.

The *action* is generated coherently with the feedback, and serves as an instruction to answering the question. Table 1 shows that for ambiguous questions, the action asks to specify the answer in detail for multiple facets, and for questions with FP, the action asks to correct the FP.

## 3.3 Answer Generation

When generating the final answer $a$, the feedback and action $(F, A)$ from the previous step act as a guideline that the language model can refer to. We concatenate $q$ and $(F, A)$ and feed it into the model with $N$ pairs of (question, feedback-action, answer) as few-shot examples within a designed prompt (See Appendix A).

## 4 Experiments and Results

**Datasets** We evaluate PreWoMe on three distinct types of questions. For misleading questions with ambiguity, we utilize ASQA (Stelmakh et al., 2022), a LFQA dataset specifically curated for answering ambiguous questions. For misleading questions with False Presupposition (FP), we use the subset of (QA)$^2$ (Kim et al., 2022) that are labeled as harboring FP. For normal questions, we use BioASQ (Krithara et al., 2023), considering that questions in BioASQ are unlikely to be ambiguous or have FP as biomedical experts have carefully curated the questions. Detailed statistics for each dataset are in Appendix C.1.

**Evaulation** For ASQA, we report Rouge-L (Lin, 2004), D-F1 (Disambiguated F1), and DR (Disambiguation-ROUGE) following Stelmakh et al. (2022). Detail of each metric is in Appendix C.2. However, rather than using the f-measure, we propose using *recall-measure* of Rouge-L instead. This is because we notice that ASQA does not provide answers to all aspects of ambiguity, and thus using the f-measure against ASQA cannot fully reflect the comprehensiveness of the generated answer, which is also discussed in Amplayo et al. (2022) (See Appendix C.3). For (QA)$^2$ and BioASQ, we report f1-measure of Rouge-1, Rouge-L, and BleuRT (Sellam et al., 2020).

**Models** We evaluate PreWoMe on two LLMs – GPT-3.5 (`gpt-3.5-turbo`) and GPT-4 (`gpt-4`) (OpenAI, 2023) – provided by OpenAI with default hyperparameters and temperature set to 0.

## 4.1 Main Experiment

We compare PreWoMe with the vanilla LLM, which involves providing only a question without

| | Method | (a) ASQA | | | (b) (QA)$^2$ | | | (c) BioASQ | | |
|---|---|---|---|---|---|---|---|---|---|---|
| | | R-L | D-F1 | DR | R-1 | R-L | B-RT | R-1 | R-L | B-RT |
| GPT-4 | Vanilla | 43.66 | 34.71 | 38.93 | 24.77 | 22.77 | 0.42 | 18.34 | 20.48 | 0.45 |
| | CoT | 49.16 | 34.59 | 41.24 | 26.12 | 24.57 | 0.43 | 20.16 | **22.42** | 0.46 |
| | Query Refinement | 49.57 | **35.32** | 41.84 | – | – | – | – | – | – |
| | Step-by-Step w. TD | – | – | – | 26.31 | 24.7 | 0.44 | – | – | – |
| | PreWoMe (Ours) | **51.04** | 35.11 | **42.34** | **30.04** | **28.30** | **0.45** | 19.85 | 22.12 | **0.48** |
| | - w/o. Presup | 42.68 | 33.41 | 37.76 | 24.67 | 22.89 | 0.42 | 17.96 | 20.23 | 0.46 |
| | - w/o. $(F,A)$ | 43.20 | 34.31 | 38.50 | 25.35 | 23.27 | 0.43 | **20.87** | 22.19 | 0.46 |
| GPT-3.5 | Vanilla | 44.83 | 30.52 | 36.99 | 24.37 | 22.49 | 0.42 | 19.65 | 22.43 | 0.46 |
| | PreWoMe | 39.58 | 28.45 | 33.55 | 26.54 | 24.80 | 0.43 | 17.72 | 20.35 | 0.46 |
| | - w. GPT-4 $(F,A)$ | **50.66** | **31.63** | **40.03** | **28.54** | **26.56** | **0.44** | **19.90** | **22.50** | **0.49** |

Table 2: Results on GPT-4 and GPT-3.5. "CoT" is Kojima et al. (2022). "Query Refinement" is proposed in Amplayo et al. (2022). "Step-by-Step w. TD" is method using task decomposition (TD) proposed in Kim et al. (2022). "w/o. Presup" is generating feedback-action *without* using presuppositions. "w/o. $(F,A)$" is using only presupposition *without* feedback and action. "w. GPT-4 $(F,A)$" is using feedback-action from GPT-4 to generate answers on GPT-3.5.

any feedback or action $(F,A)$. We use six question-answer pairs as few-shot examples ($N = 6$) for both PreWoMe and vanilla LLM. We also compare PreWoMe with Chain-of-Thought Prompting ("CoT"), following the prompt format of Kojima et al. (2022). As additional baselines, we consider two approaches that each target a single type of misleading question: Amplayo et al. (2022) proposed using refined prompts to make LLMs explicitly consider multifaceted aspects of questions to handle ambiguous questions ("Query Refinement" in Table 2). Kim et al. (2022) proposed a prompt that combine CoT (Kojima et al., 2022) with Task Decomposition (TD) prompting (Khot et al., 2022) to handle questions with FP ("Step-by-Step w. TD" in Table 2). To validate the capability of PreWoMe on misleading questions, we compare PreWoMe against these two additional baselines for each corresponding target dataset.

As shown in Table 2, we found that for GPT-4, our approach performs well on misleading questions as well as normal questions, outperforming the vanilla LLM across all datasets. Additionally, PreWoMe outperforms CoT on misleading questions (ASQA and (QA)$^2$) and yields results comparable to those of CoT for BioASQ. It is noteworthy that CoT itself demonstrates better performance than the vanilla LLM, supporting that utilizing intermediate steps improves QA performance in LLMs. In addition, PreWoMe outperforms both of the additional baselines. Especially considering that Query Refinement was optimized for ASQA and Step-by-Step w. TD was designed for (QA)$^2$, these findings suggest that our approach is versatile in handling arbitrary types of misleading questions.

For GPT-3.5, we found that the vanilla LLM shows better performance on some datasets than PreWoMe. Through manual analysis, we observed that GPT-3.5 lacks the ability to generate high-quality feedbacks and actions, compared to GPT-4 (See Appendix D). Thus we conducted experiments where we replaced feedback and action $(F,A)$ with those generated by GPT-4 and generated the answer on GPT-3.5. The results outperformed not only the vanilla GPT-3.5, but also the vanilla GPT-4, demonstrating that 1) misleading $(F,A)$ can induce noise, 2) high-quality $(F,A)$ is transferable between different models, and 3) can positively impact LLMs to the extent where GPT-3.5 outperforms GPT-4.

For further validation on different N-shot ($N = 4, 8$), we conducted additional experiments, which can be found in Appendix E.

## 4.2 Human Evaluation

For human evaluation, we randomly sample a total of 90 questions from three datasets and provided two generated answers, one from PreWoMe and one from the vanilla LLM, with the golden answer to three human evaluators. For each criterion – overall impression ($OI$), information completeness ($CP$), and correctness ($CR$) – we ask human evaluators to choose a better answer between two generated answers or mark them as a tie, inspired by Stelmakh et al. (2022). Then, we adopt a majority voting policy for each question based on the votes given by each human evaluator. Figure 3 shows that PreWoMe outperforms the vanilla LLM for OI. The notable difference between PreWoMe and vanilla LLM in CP for ASQA and CR for (QA)$^2$ suggests that PreWoMe is capable of effectively covering

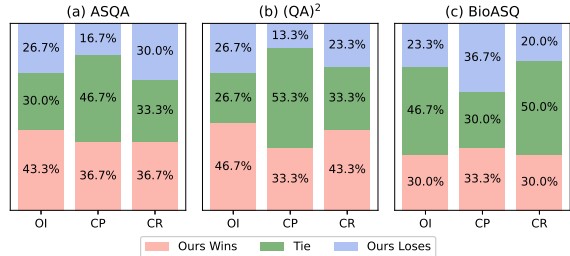

Figure 3: Human evaluation results for each dataset. We compare PreWoMewith vanilla LLM (GPT-4) over 90 questions in total.

| | | | Prediction | | Performance | |
|---|---|---|---|---|---|---|
| | | Normal | Ambig. | FP | Correct | Incorrect |
| Golden | Normal | **74.9%** | 18.5% | 6.6% | 23.19 / 21.26 | 19.03 / 18.25 |
| | Ambig. | 54.6% | **35.9%** | 9.5% | 39.70 / 34.04 | 43.65 / 41.51 |
| | FP | 11.6% | 14.0% | **74.4%** | 30.45 / 22.38 | 21.93 / 24.02 |

Table 3: *Prediction* shows distribution of predicted question type of PreWoMe using generated feedback. *Performance* indicates result of (PreWoMe / vanilla LLM) with correct and incorrect predictions.

more aspects of ambiguity and correcting FP when facing misleading questions. The prompt used for human evaluation is provided in Appendix F.

## 4.3 Ablation Studies

For ablation studies, we first conduct analysis on the LLM's capability of handling each intermediate step, then verify the importance of each intermediate step within PreWoMe.

**Presupposition Analysis** To analyze the LLM's capability of generating the correct presuppositions, we randomly sample 150 questions and manually annotate whether the corresponding 395 generated presuppositions were indeed assumed by the question. The accuracy of presuppositions was extremely high (391 correct out of 395 total $\approx$ 99%), demonstrating GPT-4's capability of extracting accurate presuppositions.

**Feedback Analysis** As mentioned in subsection 3.2, feedback works as a classifier that determines the type of the given question. We extract the question type predicted by the feedback using a rule-based approach and report the distribution in Table 3 (*Prediction*). Feedback effectively detects normal questions and questions with FP (74.9% and 74.4%). However, ambiguous questions have low detect ratio, consistent with Min et al. (2020). This indicates that there is more room to optimize for detecting ambiguity in the question, which we leave for future work.

We also report the final performance of Pre-WoMe and vanilla LLM when given correctly and incorrectly classifying feedback (Table 3 *Performace*). Specifically, we report D-F1 for ASQA and Rouge-L for BIOASQ and $(QA)^2$ of PreWoMe and vanilla GPT-4. It can be seen that regardless of the question type, performance gains increase when feedback correctly classifies the question type.

**Importance of Presuppositions** Do presuppositions really contribute to generating better feedback and action? We explore this question by omitting presupposition extraction step and generating feedback and action with GPT-4 just based on the question. As shown in Table 2 (w/o. Presup), PreWoMe without presuppositions extraction shows poor performance, highlighting the role of presuppositions as working memory. One interesting observation is that the performance of PreWoMe without presupposition is worse than the vanilla LLM, indicating that misleading $(F,A)$ becomes a noise, consistent with findings in subsection 4.1.

**Importance of Feedback, Action** Similarly, we also explore the importance of Feedback and Action. We omit the feedback and action generation step and just feed the questions with the presuppositions that are generated in the previous step into GPT-4. Table 2 (w/o. $(F,A)$) shows that using presuppositions without feedback/action generation step shows lower performance compared to the full PreWoMe for misleading questions. This result demonstrates the importance of feedback/action, and also that generating presuppositions itself is not sufficient to handle misleading questions.

## 5 Conclusion

We propose PreWoMe, a new approach that handles any type of question in a unified way by exploiting **presuppositions as working memory**. PreWoMe uses intermediate steps to guide the model to generate presuppositions along with useful feedback and action for itself. Our experiments show that PreWoMe boosts the performance of LLMs on different types of questions without parameter updates. This is achieved even though the model is given no information about the type of incoming question, demonstrating that PreWoMe is effective for a real-world setting and will be a significant stepping stone to future works tackling real-world QA tasks.

## Limitations

As we discussed in subsection 4.3, we note that misconstructed feedback and action can be a noise for LLMs, which leads them to generate incorrect answers in the answer generation step. This phenomenon, similar to hallucination snowballing, was also pointed out by Zhang et al. (2023). Future work may improve PreWoMe by exploring how to prevent hallucination snowballing.

Also, our work focuses on closed-book question-answering and explores performance only on closed-book question-answering tasks. Expanding PreWoMe's approach to tasks such as question-answering with context, text summarization, etc will be also an avenue of future work.

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

## A Designed Prompts and Few-shots

Table 4 shows specific designed prompts and the structure of few-shots used in subsection 4.1.

## B Normal Question Example

Like Table 1, we give an example of assumptions, feedback, and action on a *normal question* (question without little to no imperfections). Numbering $A$ in the list of presuppositions stands for the additional presupposition $p_A$, which is manually added by post-processing extracted presuppositions as explained in subsection 3.1.

It can be seen in Table 5 that the *feedback* informs that the question does not contain any FP. In this case, the *action* functions as an elaboration of the question.

## C Dataset

### C.1 Dataset Statistics

**ASQA** ASQA includes train, development, and test split each of which contains 4,353, 948, and 1,015 questions. For evaluation, we use the development split of ASQA because the test split of ASQA is not open publicly.

**(QA)$^2$** (QA)$^2$ consists of adaptation and evaluation splits, each of which contains 32 and 570 questions, respectively. For each split, half of them are questions that include FP, and the other half are questions that do not. For evaluation, we use half of the evaluation split – only questions that contain the FP.

**BioASQ** BioASQ consists of two distinct tasks (Task A, and Task B) and periodically releases datasets for each task. We use BioASQ9B, which is task B released in 2021, and concatenated the 5 batches (9B1_golden, 9B2_golden, ···, and 9B5_golden) in the test split for evaluation. We regard *ideal_answer* in the dataset as a gold reference. Following Stelmakh et al. (2022), we compare predictions against all answers in *ideal_answer* and report the maximum score. The concatenated test split includes 497 question-answer pairs.

### C.2 ASQA Metric

Stelmakh et al. (2022) proposed several metrics for evaluating ASQA.

**ROUGE-L** Stelmakh et al. (2022) used a f-measure of ROUGE-L (Lin, 2004) score which is a metric for evaluating generated text.

**Disambig-F1** Stelmakh et al. (2022) used Disambig-F1 score to evaluate the *informativeness* of generated text. The ambiguous question $q$ can be disambiguated to multiple pairs of disambiguous questions and short answers $(q_i, a_i)$. For each ambiguous question $q$, they feed system-generated long-form answer as a context, along with each disambiguated question ($q_i$) to a Roberta Model (Liu et al., 2019) that was pretrained on SQUADv2 (Rajpurkar et al., 2018) to predict a short answer. Then they calculated the token-level F1 score between the predicted short answer from Roberta and the gold short answer $a_i$. Then, Disambig-F1 score of a single ambiguous question $q$ is given by averaging

the calculated F1 score of all corresponding disambiguated questions. The final Disambig-F1 score is calculated by averaging Disambig-F1 scores of all questions in the split.

**Overall DR Score** Stelmakh et al. (2022) proposed a novel metric for evaluating ASQA: DR Score. Specifically, the DR Score is given by calculating the geometric mean of Rouge-L and Disambig F1 as expressed as follows:

$$DR = \sqrt{(Disambig - F1) * Rouge}.$$

### C.3 Recall-measure Rouge-L

In this part, we explain the reason for using the *recall-measure* of Rouge-L instead of the f-measure with an actual example. The ASQA dataset (Stelmakh et al., 2022) consists of ambiguous questions and their disambiguated versions with corresponding answers. However, we have noticed that the disambiguated question-answer pairs in ASQA do not always cover all aspects of ambiguity inherent in the ambiguous question. Thus, even though PreWoMe generates an answer for the ambiguous question with factually correct disambiguations, if that disambiguation and answer were not provided by ASQA, using the f1-measure will penalize the factually correct answers just because they were not part of the ASQA dataset.

While we *do not give higher scores* for our factually correct disambiguated answers, we thought it is fair to use a recall-measure to stop getting scores *deducted* because of them. Table 6 gives an example. The generated answer of PreWoMe includes an explanation of "Stuck in the Middle" by Tai Verdes and "Stuck in the Middle" by Boys Like Girls which are reasonable answers to the ambiguous question. However, as ASQA does not include any disambiguated question-answer pairs that cover such aspects, using the f1-measure will give a even lower score compared to the prediction without those explanations.

### D Quality of Feedback and Action

As discussed in subsection 4.1 and 4.3, poorly generated feedback and action make LLMs underperform. Table 7 shows some examples of such cases. In Table 7 (a), it can be seen that the generated action is already itself the answer. This makes GPT-3.5 generate the answer by merely copying the action, which does not cover the ambiguity of the question. However, we can notice that GPT-3.5 can, in fact, discern the ambiguity of the ques-

tion, by observing that the ambiguity is considered in the answer of the vanilla GPT-3.5. In Table 7 (b), GPT-3.5 was misled by the generated action. Specifically, GPT-3.5 answered that there was no fight between Muhammad Ali and Michael Dokes which is in fact not true. The last row of (b) indicates that GPT-3.5 has knowledge about the fight between two boxers, which was not exploited due to a faulty action.

### E Experiment on different number of few-shots

In this part, we show additional experiments conducted using $N = 4, 8$ numbers of few-shot examples (Table 8, Table 9). The overall trend is consistent with Table 2, demonstrating that PreWoMe is robust to different numbers of few-shot examples.

### F Human Evaluation Prompt

In this part, we release the prompt that are given to human evaluators in Table 10.

| | Prompt | You are a helpful assistant that analyzes the following question. Your task is to extract assumptions implicit in a given question. You must notice that considering the intention of the question will be helpful to extract a hidden assumption of the given question. |
|---|---|---|
| | Few-shot | Question : When did the great depression began before world war 1?
Presuppositions :
(1) There was a period called the Great Depression.
(2) There was a conflict called World War 1.
(3) The Great Depression began before World War 1. |

Step 1: Presupposition Extraction. $p_A$ is added in the post-processing step.

| | Prompt | You are a helpful assistant that provides a feedback on the question and a guideline for answering the question. You will be given a question and the assumptions that are implicit in the question. Your task is to first, provide feedback on the question based on whether it contains any false assumption and then provide a guideline for answering the question. |
|---|---|---|
| | Few-shot | Question : When did the great depression began before world war 1?
Presuppositions :
(1) There was a period called the Great Depression.
(2) There was a conflict called World War 1.
(3) The Great Depression began before World War 1.
(4) There is a clear and single answer to the question.
Feedback : The question contains a false assumption that the Great Depression began before World War 1.
Action : Correct the false assumptions that the Great Depression began before World War 1 and respond based on the corrected assumption. |

Step 2: Feedback and Action generation

| Method | PreWoMe | Vanilla |
|---|---|---|
| Prompt | You are a helpful assistant that answers the given question. The feedback on the question and action needed for answering the question will also be given. Write a coherent and detailed answer to the question below. | You are a helpful assistant that answers the given question. Write a coherent and detailed answer to the question below. |
| Few-shot | Question : When did the great depression began before world war 1?
Feedback : The question contains a false assumption that the Great Depression began before World War 1.
Action : Correct the false assumptions that the Great Depression began before World War 1 and respond based on the corrected assumption.
The Great Depression did not begin before World War I. The Great Depression began in 1929, which was after World War I (1914-1918). | Question : When did the great depression began before world war 1?
The Great Depression did not begin before World War I. The Great Depression began in 1929, which was after World War I (1914-1918). |

Step 3: Answer generation.

Table 4: Designed prompts and one few-shot example used in each step. PreWoMe takes all three steps, while the vanilla LLM only takes the Answer Generation step. In our subsection 4.1, we concat $N = 6$ few shots that each have the structure shown in the table above.

| Type | Normal Questions |
|---|---|
| Question | Is the TFR1 gene dispensable for erythropoiesis? |
| Presuppositions | 1) There is a gene called TFR1.
2) It is possible for a gene to be dispensable or essential for a biological process.
A) There is a clear and single answer to the question. |
| Feedback | **The question does not contain any false presuppositions. All the presuppositions are valid.** |
| Action | Answer the question by discussing the role of the TFR1 gene in erythropoiesis and whether it is essential or dispensable for the process. Provide evidence from scientific research to support the answer. |

Table 5: Examples of normal question from (Krithara et al., 2023) and the results of intermediate steps throughout our pipeline.

| Ambiguous Question |
| --- |
| Who sings the song for stuck in the middle? |

| Disambiguated Question-Answer Pairs |
| --- |
| **Q1**) Which band sings the 1973 song "Stuck in the Middle with You"? -> **A1**) Stealers Wheel |
| **Q2**) Who sings the lead vocals in the 1973 song "Stuck in the Middle with You"? -> **A2**) Gerry Rafferty |
| **Q3**) Who sings harmony in the 1973 song "Stuck in the Middle with You"? -> **A3**) Joe Egan |
| **Q4**) Who sings the 2004 song "Stuck in the Middle"? -> **A4**) Clea |
| **Q5**) Who sings the theme song for the TV show Stuck in the Middle? -> **A5**) Sonus |

| Generated answer on Ambiguous Question |
| --- |
| There are multiple songs titled "Stuck in the Middle" sung by different artists. Some of them include: |
| 1. "Stuck in the Middle with You" by **Stealers Wheel**: This song was released in 1972 and was written by Gerry Rafferty and Joe Egan. It became a hit single and is often associated with the 1992 Quentin Tarantino film, Reservoir Dogs. |
| 2. "Stuck in the Middle" by **Tai Verdes**: This song was released in 2020 and is part of Tai Verdes' debut album, "TV." The song gained popularity on social media platforms like TikTok. |
| 3. "Stuck in the Middle" by **Boys Like Girls**: This song is from their 2006 self-titled debut album. The band is an American pop-rock group formed in 2005. |
| These are just a few examples, and there may be other songs with the same or similar titles sung by different artists. |

Table 6: Example from ASQA (Stelmakh et al., 2022) that shows why using a f1-measure for Rouge-L is inappropriate. The disambiguated question-answer pairs do not include all factually correct answers (Tai Verdes, Boys Like Girls), therefore penalizing generated answers even though they are more comprehensive and complete.

| **Type** | **Ambiguous Questions** |
| --- | --- |
| Question | Who played zordon in the original power rangers? |
| Generated Feedback | The question contains a true assumption. |
| Generated Action | The actor who played Zordon in the original Power Rangers series is David J. Fielding. |
| Generated Answer | The actor who played Zordon in the original Power Rangers series is **David J. Fielding**. |
| Answer by vanilla GPT-3.5 | In the original Mighty Morphin Power Rangers television series, Zordon was voiced by actor **David J. Fielding**. However, Zordon's face was portrayed by actor **Robert L. Manahan**, who was credited as "Zordon's Voice" in the show's credits. |
| Golden Answer | In the 1993 American superhero television series Mighty Morphin Power Rangers, fictional character Zordon is initially voiced and portrayed by David Fielding, and later voiced by Robert L. Manahan. In the 1995 American superhero film Mighty Morphin Power Rangers: The Movie, Zordon is portrayed by Nicholas Bell. |

(a) Example of poorly constructed feedback and action for the ambiguous question.

| **Type** | **Ambiguous Questions** |
| --- | --- |
| Question | Who won muhammad ali vs michael dokes? |
| Generated Feedback | The question contains a false assumption that there was a fight between Muhammad Ali and Michael Dokes. |
| Generated Action | Correct the false assumption that there was a fight between Muhammad Ali and Michael Dokes and respond based on the corrected assumption. |
| Generated Answer | Muhammad Ali and Michael Dokes never fought against each other in a professional boxing match. Therefore, there is no winner to report. |
| Answer by vanilla GPT-3.5 | Muhammad Ali won the fight against Michael Dokes. The fight took place on September 10, 1977, at the The Forum in Inglewood, California. Ali won the fight by a unanimous decision after 10 rounds. |
| Golden Answer | In April 1977, Muhammad Ali fought two exhibitions of three rounds each against Jody Ballard and future World Heavyweight champion Michael Dokes, both to a No-Decision. |

(b) Example of poorly constructed feedback and action for the question that includes FP.

Table 7: Examples of feedback and action that mislead LLMs.

| Model | Method | (a) ASQA | | | (b) $(QA)^2$ | | | (c) BioASQ | | |
|---|---|---|---|---|---|---|---|---|---|---|
| | | Rouge-L | D-F1 | DR | Rouge-1 | Rouge-L | BleuRT | Rouge-1 | Rouge-L | BleuRT |
| GPT-4 | Vanilla | 41.63 | 34.61 | 37.96 | 24.54 | 22.42 | 0.42 | 18.53 | 20.68 | 0.45 |
| | PreWoMe | **45.68** | **35.76** | **40.42** | **27.28** | **25.33** | **0.44** | **18.96** | **21.2** | **0.47** |
| GPT-3.5 | Vanilla | 43.45 | 30.74 | 36.55 | 23.99 | 22.13 | 0.41 | **19.94** | **22.35** | 0.46 |
| | PreWoMe | 41.22 | 28.36 | 34.19 | 26.78 | 24.93 | 0.43 | 17.31 | 19.79 | 0.46 |
| | - w. GPT-4 $(F,A)$ | **45.83** | **30.79** | **37.57** | **27.13** | **25.42** | **0.44** | 18.96 | 21.82 | **0.47** |

Table 8: Result of PreWoMe and vanilla GPT-4 and GPT-3.5 using $N = 4$ few-shot examples. The last row (w. GPT-4 $(F,A)$) refers to the system that generates answers on GPT-3.5 using feedback and action generated by GPT-4.

| Model | Method | (a) ASQA | | | (b) $(QA)^2$ | | | (c) BioASQ | | |
|---|---|---|---|---|---|---|---|---|---|---|
| | | Rouge-L | D-F1 | DR | Rouge-1 | Rouge-L | BleuRT | Rouge-1 | Rouge-L | BleuRT |
| GPT-4 | Vanilla | **42.51** | 34.98 | 38.56 | 25.72 | 23.68 | 0.43 | 18.82 | 20.95 | 0.45 |
| | PreWoMe | 42.07 | **35.82** | **38.82** | **33.53** | **31.68** | **0.46** | **21.55** | **24.15** | **0.46** |
| GPT-3.5 | Vanilla | 41.94 | 30.28 | 35.64 | 25.46 | 23.21 | 0.42 | 20.68 | 23.26 | 0.46 |
| | PreWoMe | 40.88 | 28.73 | 34.27 | 28.60 | 25.67 | 0.44 | 19.61 | 22.10 | 0.46 |
| | - w. GPT-4 $(F,A)$ | **45.39** | **31.38** | **37.74** | **30.38** | **28.11** | **0.45** | **21.65** | **24.35** | 0.46 |

Table 9: Result of PreWoMe and vanilla GPT-4 and GPT-3.5 using $N = 8$ few-shot examples. The last row (w. GPT-4 $(F,A)$) refers to the system that generates answers on GPT-3.5 using feedback and action generated by GPT-4.

In this task, you will be shown one question and its golden answer, and two generated answers from different systems.

Your goal is to evaluate which answer is better for each of the three criteria (Overall Impression, Completeness, Correctness) by referring to the given golden answer.

For the Overall Impression criterion, your job is to select an answer (or mark it as a tie) that gives you more satisfaction. Fluency, correctness, consistency, sufficiency of information, or even formatting can be considered as a factor.

For the Completeness criterion, your job is to select an answer (or mark it as a tie) considering whether the answer provides enough information. You can refer to the given golden answer for the evaluation.

For the Correctness criterion, your job is to select an answer (or mark it as a tie) considering whether the answer does not include any hallucination (factually wrong information).

You can refer to the given golden answer for the evaluation.

Table 10: Prompt given to human evaluator