# OpenReview forum: "PreWoMe: Exploiting Presuppositions as Working Memory for Long Form Question Answering"
_EMNLP/2023/Conference — EMNLP 2023 Main_

### Official Review · Reviewer_QSHM · 2023-07-30

**Soundness:** 4

**Excitement:**

3: Ambivalent: It has merits (e.g., it reports state-of-the-art results, the idea is nice), but there are key weaknesses (e.g., it describes incremental work), and it can significantly benefit from another round of revision. However, I won't object to accepting it if my co-reviewers champion it.

**Paper Topic And Main Contributions:**

This paper proposes a multi-step prompting method for closed-book long-form QA, which includes three steps: extracting presupposition statements from the question; generating feedback and action given the extracted presupposition and finally generating the answer given the question, feedback and action. Experiments are conducted on three datasets, covering ambiguous questions, questions with false-presuppositions and a biomedical dataset. Automatic evaluation and human evaluations show that the proposed method outperforms the baseline few-shot approach.

**Questions For The Authors:**

* In line 208 – does “few-shot examples (N=6)” for PreWoMe mean having 6 examples for all the steps, or just the final answer generation step?

* What are the details for human evaluation?  What’s the instruction, who are the evaluators and how much do they agree with each other on the choices?

**Reasons To Accept:**

* This paper proposes a new prompting approach for generating long-form answers.
* The author conducted both automatic and human evaluation on multiple datasets (ASQA, QA^2 and BioASQ), covering different types of questions.
* The paper is generally well-written and easy to follow.

**Reasons To Reject:**

My major concern is the lack of baseline and ablation study here.

* Baseline: the authors only compared against a few-shot baselines. The query refinement method proposed in [1] should be a reasonable baseline to compare against, at least for the ASQA dataset. Another baseline is the step-by-step baseline from [2] (table 9). As these two methods are both one-step (whereas the model proposed involves prompting the model three times), it’d be interesting to analyze the computational cost v.s. performance tradeoff.

* Ablation: results w/o presupposition and having different numbers of few-shot examples are reported. However as there are two steps before generating the final answer, it’d be interesting to see if just having the list of presuppositions and a general prompt to ask the model to generate the final answer is enough, or the second step of prompting the LLM to get a specific feedback for the presupposition is needed.

[1] Reinald Kim Amplayo, Kellie Webster, Michael Collins, Dipanjan Das, and Shashi Narayan. 2022. Query refinement prompts for closed-book long-form question answering.

[2] Najoung Kim, Phu Mon Htut, Samuel R Bowman, and Jackson Petty. 2022. Θ 2: Question answering with questionable assumptions.

**Reproducibility:**

4: Could mostly reproduce the results, but there may be some variation because of sample variance or minor variations in their interpretation of the protocol or method.

**Reviewer Confidence:**

4: Quite sure. I tried to check the important points carefully. It's unlikely, though conceivable, that I missed something that should affect my ratings.

---

> ### Author Rebuttal · Authors · 2023-08-29
>
> Dear Reviewer QSHM.
>
> We sincerely appreciate all your comments and suggestions. We respond to specific review comments below.
>
> &nbsp;
>
> > **Weakness #1**: Baseline: The authors only compared against few-shot baselines. The query refinement method proposed in [1] should be a reasonable baseline to compare against, at least for the ASQA. Another baseline is the step-by-step baseline from [2] (table 9). As these two methods are both one-step (whereas the model proposed involves prompting the model three times), it’d be interesting to analyze the computational cost v.s. performance tradeoff.
>
> **Answer**:
>
> We conducted the experiment with two additional baselines. The results are as follows:
>
> | ASQA                        | Rouge-L | Disambig-F1 | Disambig-Rouge |
> | --------------------------- | ------- | ----------------- | -------------------- |
> | PreWoMe                     | **51.04**   | 35.11             | **42.34**                |
> | Query Refinement Method [1] | 49.57   | **35.32**             | 41.84                |
> | Step-by-step [2]            | 41.62   | 33.18             | 37.16                |
> | Baseline                        | 41.63  | 34.61             | 37.96                |
>
>
> | QA$^2$                      | Rouge-1 | Rouge-L | BleuRT |
> | --------------------------- | ------- | ----------------- | -------------------- |
> | PreWoMe                     | **30.04**   | **28.3**    | **0.45**   |
> | Query Refinement Method [1] | 25.52   | 23.63   | 0.41              |
> | Step-by-step [2]            | 26.31   | 24.7    | 0.44   |
> | Baseline                        | 24.54  | 22.42  | 0.42   |
>
> The results show that PreWoMe outperforms both of the additional baselines, even though [1] was specifically designed for ASQA and [2] was targeted for QA$^2$. This demonstrates that PreWoMe is effective in solving arbitrary misleading questions.
>
> Next, we also compare the computation costs for outputs of PreWoMe and baselines.  As the computation costs of autoregressive LMs are linear to their output tokens, we reported the average length of output tokens for each system in the table below.  Specifically, we analyze the cost of generating intermediate steps before generating the final answer (i.e. presupposition/feedback/action steps for PreWoMe, and intermediate steps for [1] and [2],
> excluding the tokens of final answers).
>
> | ASQA         |  # Avg Output Token
> | ------------ |  -----------
> | PreWoMe      |  106.78
> | Query Refinement Method [1]          |  113.86
> | Step-by-step [2]          |  16.92
>
> | QA$^2$       | # Avg Output Token |
> | ------------ |  -------
> | PreWoMe | 107.75
> | Query Refinement Method [1]          | 151.38
> | Step-by-step [2]          | 84.21
>
> Surprisingly, even though PreWoMe includes multiple steps for generating the answer, the cost of PreWoMe is lower than that of Query Refinement Method [1] for both ASQA and QA$^2$. Nevertheless, the performance of [1] was lower than our PreWoMe.
>
> Step-by-step method [2] demands significantly lower costs compared to others. In Particular, the cost was minimal for the ASQA dataset. We observed the step-by-step method [2] does not tend to generate intermediate steps such as rationale on ASQA. Also, the performance of [2] for ASQA is even lower than the baseline.
>
> &nbsp;
>
> > **Weakness #2**: Results w/o presupposition and having different numbers of few-shot examples are reported. However as there are two steps before generating the final answer, it’d be interesting to see if just having the list of presuppositions and a general prompt to ask the model to generate the final answer is enough, or the second step of prompting the LLM to get a specific feedback for the presupposition is needed.
>
> **Answer**:
>
> To check the effect of feedback/action generation step on the final answer, we experimented with PreWoMe by just generating the presuppositions, and skipping the feedback/action generation step (denoted as *w/o. F/A* below). The experiment results “without presuppositions and only feedback/action generation” are shown in Table 2 (denoted as *w/o. Presup*) of our paper.
>
> | ASQA         | Rouge-L | Disambig-F1 | Disambig-Rouge |
> | ------------ | ------- | ----------- | -------------- |
> | Full PreWoMe | **51.04**   | **35.11**       | **42.34**          |
> | w/o. F/A  | 43.2    | 34.31       | 38.5           |
> | w/o. Presup     | 42.68   | 33.51       | 37.76          |
>
> | QA$^2$       | Rouge-1 | Rouge-L | BleuRT |
> | ------------ | ------- | ------- | ------ |
> | Full PreWoMe | **30.04**   | **28.3**    | **0.45**   |
> | w/o. F/A  | 25.35   | 23.27   | 0.43   |
> | w/o. Presup     | 24.67   | 22.89   | 0.42   |
>
> | BioASQ       | Rouge-1 | Rouge-L | BleuRT |
> | ------------ | ------- | ------- | ------ |
> | Full PreWoMe | 19.85   | 22.12   | **0.48**   |
> | w/o. F/A  | **20.87**   | **22.19**   | 0.46   |
> | w/o. Presup     | 17.96   | 20.23   | 0.46   |
>
> It can be seen that using only presuppositions without feedback/action generation step shows lower performance compared to the full PreWoMe for misleading questions. However, it can be seen that it performs slightly better than PreWoMe without presuppositions (that is, just using feedback/action). As mentioned in our paper, this quantitatively demonstrates the importance of presuppositions, and also that incorrect feedback/action can potentially induce noise.
>
> &nbsp;
>
> > **Question #1**: In line 208 – does “few-shot examples (N=6)” for PreWoMe mean having 6 examples for all the steps, or just the final answer generation step?
>
> **Answer**:
>
> We have used 6 shots throughout all steps of PreWoMe.
> To elaborate, PreWoMe is designed for a realistic setting, where input question types are unknown. Thus, we use few shots sampled from multiple datasets, and use those same few shots regardless of the type of input question or datasets.
> An example of how the few shots for each step are constructed is illustrated in Table 4 of our paper’s appendix.
>
> &nbsp;
>
> > **Question #2**: What are the details for human evaluation? What’s the instruction, who are the evaluators and how much do they agree with each other on the choices?
>
> **Answer:**
>
> We hired three undergraduate students as human evaluators.
> For evaluation, we provided the evaluators with a question, a golden answer, and two generated answers (one from the baseline, and one from PreWoMe). Then, we asked them to select one of the two answers based on three criteria - overall impression (OI), information completeness (CP), and correctness (CI) - as mentioned in the paper.
> The order of two generated answers was randomly shuffled for every question.
>
> This is the prompt that was given to the human evaluators;
>
> *In this task, you will be shown one question and its golden answer, and two generated answers from different systems. Your goal is to evaluate which answer is better for each of the three criteria (Overall Impression, Completeness, Correctness) by referring to the given golden answer. For the Overall Impression criterion, your job is to select an answer (or mark it as a tie) that gives you more satisfaction. Fluency, correctness, consistency, sufficiency of information, or even formatting can be considered as a factor. For the Completeness criterion, your job is to select an answer (or mark it as a tie) considering whether the answer provides enough information. You can refer to the given golden answer for the evaluation. For the Correctness criterion, your job is to select an answer (or mark it as a tie) considering whether the answer does not include any hallucination (factually wrong information). You can refer to the given golden answer for the evaluation.*
>
> The table below reports the Fleiss Kappa score, which shows how much the human evaluators agree with each other.
> It can be seen that all scores range between 0.4 and 0.6, which is interpreted as having a ‘moderate agreement’ between the evaluators (based on the widely used interpretation table proposed in [3]).
>
> |    | ASQA | QA$^2$ | BioASQ |
> | -- | ---- | ------ | ------ |
> | Overall Impression | 0.56 | 0.55   | 0.48   |
> | Completeness | 0.57 | 0.53   | 0.4    |
> | Correctness | 0.48 | 0.54   | 0.5    |
>
>
> ----------------------
>
> **Reference**
>
> [1] Amplayo, Reinald Kim, et al. "Query Refinement Prompts for Closed-Book Long-Form Question Answering." arXiv preprint arXiv:2210.17525 (2022).
>
> [2] Kim, Najoung, et al. "QA$^2$: Question Answering with Questionable Assumptions." arXiv preprint arXiv:2212.10003 (2022).
>
> [3] Landis, J. Richard, and Gary G. Koch. "The measurement of observer agreement for categorical data." biometrics (1977): 159-174.

---

### Official Review · Reviewer_4fGt · 2023-08-03

**Soundness:** 4

**Excitement:**

4: Strong: This paper deepens the understanding of some phenomenon or lowers the barriers to an existing research direction.

**Paper Topic And Main Contributions:**

This paper presents a new approach for misleading factoid questions (i.e., with false presuppositions or are ambiguous). In specific, the method extracts presuppositions from the question using LLMs, and prompts LLMs to generate [feedbacks, actions] based on the question and presupposition (as working memory). LLMs then are used to generate the answer based on the [question, feedback, action] triplets. Experiments are conducted with misleading questions and answerable factoid questions. Results show the new approach outperforms the baseline by considering only the question.

**Reasons To Accept:**

1. The method is unified to both ambiguous questions and FP questions, which is novel.
2. The idea of extracting presupposition from the question and generating answers by leveraging it is technically sound.
3. The paper is well-written. Experimental details are explained clearly.

**Reasons To Reject:**

1. There's no evaluation or analysis provided for the intermediate generated presupposition, feedback, and action. It would be interesting to see LLM's capabilities in handling such subtasks. Besides, if the intermediate feedback and action are wrong or misleading, there is no point to feed them to the answer generation step.


**Reproducibility:**

5: Could easily reproduce the results.

**Reviewer Confidence:**

5: Positive that my evaluation is correct. I read the paper very carefully and I am very familiar with related work.

---

> ### Author Rebuttal · Authors · 2023-08-29
>
> Dear Reviewer 4fGt.
>
> We sincerely appreciate all your comments and suggestions. We respond to specific review comments below.
>
> &nbsp;
>
> > **Weakness #1**: There's no evaluation or analysis provided for the intermediate generated presupposition, feedback, and action. It would be interesting to see LLM's capabilities in handling such subtasks. Besides, if the intermediate feedback and action are wrong or misleading, there is no point to feed them to the answer generation step.
>
> **Answer**:
>
> To check the LLM's capabilities in handling subtasks, we evaluated the performance of intermediate steps.
>
> *First*, we analyze the accuracy of the generated presuppositions.
> We randomly sampled 150 questions and manually annotated whether the corresponding 395 generated presuppositions are correct, or not. The accuracy of presuppositions was extremely high (Correct 391/total 395, ~99%) demonstrating GPT-4’s capability of extracting accurate presuppositions.
>
> *Second*, as mentioned in our paper, feedback can function as a classifier that predicts the type of input question. The table below reports the performance of the feedback in classifying the input question, which is also reported in our paper (Table 3).
>
> |               | Prediction.Ideal | Prediction.Ambig | Prediction.FP |
> | ------------- | ---------------- | ---------------- | ------------- |
> | Golden.BioASQ | **74.9%**            | 18.5%            | 6.6%          |
> | Golden.ASQA   | 54.6%            | **35.9%**            | 9.5%          |
> | Golden.QA$^2$ | 11.6%            | 14.0%            | **74.4%**         |
>
> Finally, we agree that feeding incorrect feedbacks to the answer generation step may induce noise, which is also mentioned in our limitation section.
> To analyze this, we observed the accuracy on correct and incorrect feedbacks (generated by PreWoMe). For BioASQ and QA$^2$, we report Rouge-L, and for ASQA we report Disambiguation-Rouge.
>
> | | Correct Feed (Ours / Base) | Incorrect Feed (Ours / Base) |
> | ------- | ---------------------- | ------------------------ |
> | BioASQ  | 23.19 / 21.26   | 19.03 / 18.25 |
> | ASQA    | 39.70 / 34.04  | 43.65 / 41.51    |
> | QA$^2$  | 30.45 / 22.38  | 21.93 / 24.02    |
>
> It is noticeable that PreWoMe tends to show higher performances especially when it successfully generates correct feedbacks.

---

### Official Review · Reviewer_M53a · 2023-08-06

**Soundness:** 3

**Excitement:**

3: Ambivalent: It has merits (e.g., it reports state-of-the-art results, the idea is nice), but there are key weaknesses (e.g., it describes incremental work), and it can significantly benefit from another round of revision. However, I won't object to accepting it if my co-reviewers champion it.

**Paper Topic And Main Contributions:**

This work proposes to treat ambiguous questions and questions with false presuppositions in a unified manner, specifically by first using a LM to generate any presuppositions within the question, and manually appending to the presuppositions: “There is a clear and single answer to the question.” In the next stage, the LM is expected to generate feedback on which presupposition is correct/incorrect (if the manually appended presupposition is deemed incorrect, the question is deemed ambiguous), followed by an action describing how best to answer the question. Experiments on three domains demonstrated the empirical effectiveness of this method over a vanilla LM baseline that simply generates answers to questions.

**Questions For The Authors:**

1. Qualitatively, how accurate are the generated presuppositions? Is there any correlation between correctly generated presuppositions and answer quality?
2. This method seems rather tailored to false presupposition detection, it is unclear to me intuitively whether and why it helps with ambiguity detection. Do the ambiguities usually have to do with one or more of the presuppositions in the question? Can this be quantified in some way, or clarified qualitatively through an example?

**Reasons To Accept:**

1. Method is simple and works empirically well when using GPT-4.
2. Human studies were run that confirm that the answers generated by PreWoMe were of higher quality than the answers generated by the baseline.
3. Analysis highlights the importance of generating presuppositions.

**Reasons To Reject:**

1. The method is somewhat similar to chain-of-thought reasoning, albeit it provides more structure to the reasoning chain than prior work. This begs the question – how does PreWoMe compare with allowing LMs to flexibly use their own “working memory” in any way they see fit (e.g. how does it compare with a “think step-by-step” baseline)?
2. It seems from the paper that all the questions evaluated in ASQA are ambiguous, and all the questions evaluated in (QA)^2 have a false presupposition. This seems to be a somewhat contrived evaluation setup that presumably is unable to test for the ability of the feedback stage to properly identify when an ambiguity or false presupposition is present or absent (the model is rewarded for always saying that the question is ambiguous / contains a false presupposition).
    1. In the n-shot demonstration setup, how are the n in-context examples sampled? If they are sampled from the same dataset, then the model can effectively “cheat” because it will only ever see in-context examples of ambiguous questions (if the test sample is ambiguous) or questions with false presuppositions (if the test sample has a false presupposition).
3. The claim that PreWoMe outperforms GPT4 on ideal questions (L211-212) is also a bit suspect (or at least requires more explanation), given that the method does not substantially differ from baseline question answering in this setting, and the empirical differences are not large.

**Reproducibility:**

4: Could mostly reproduce the results, but there may be some variation because of sample variance or minor variations in their interpretation of the protocol or method.

**Reviewer Confidence:**

4: Quite sure. I tried to check the important points carefully. It's unlikely, though conceivable, that I missed something that should affect my ratings.

---

> ### Author Rebuttal · Authors · 2023-08-29
>
> Dear Reviewer M53a.
>
> We sincerely appreciate all your comments and suggestions. We respond to specific review comments below.
>
> &nbsp;
>
> > **Weakness #1**: The method is somewhat similar to chain-of-thought reasoning, albeit it provides more structure to the reasoning chain than prior work. This begs the question – how does PreWoMe compare with allowing LMs to flexibly use their own “working memory” in any way they see fit (e.g. how does it compare with a “think step-by-step” baseline)?
>
> **Answer**:
>
> To compare PreWoMe with another baseline using working memory, we conducted additional experiments on GPT-4 with Chain of Thought (CoT) prompting [1].
>
> | ASQA         | Rouge-L | Disambig-F1 | Disambig-Rouge |
> | ------------ | ------- | ----------- | -------------- |
> | Baseline     | 43.66    | 34.71       | 38.93          |
> | CoT          | 49.16    | 34.59       | 41.24          |
> | PreWoMe      | **51.04**| **35.11**   | **42.34**      |
>
> | QA$^2$         | Rouge-1 | Rouge-L | BleuRT  |
> | ------------ | -------  | -------  | ------  |
> | Baseline     | 24.77    | 22.77    | 0.42    |
> | CoT          | 26.12    | 24.57    | 0.43    |
> | PreWoMe      | **30.04**| **28.3** | **0.45**|
>
> | BioASQ         | Rouge-1 | Rouge-L | BleuRT |
> | ------------ | ------- | ------- | ------ |
> | Baseline | 18.34   | 20.48    | 0.45   |
> | CoT      | **20.16**   | **22.42**   | 0.46   |
> | PreWoME  | 19.85   | 22.12   | **0.48**   |
>
>
> The result shows that even though CoT boosts the baseline performance, PreWoMe outperforms CoT for misleading questions and shows comparable results on ideal questions.
>
> We claim PreWoMe and CoT have different strengths.
> CoT has been demonstrated to effectively improve the reasoning abilities of LMs, such as mathematical and logical reasoning.
> Meanwhile, compared to CoT which does not consider the question's presuppositions, PreWoMe exploits presupposition, feedback, and action as working memory, and this makes LMs much more robust for open-ended QA in the wild. This is also substantiated by the above results.
>
> &nbsp;
>
> > **Weakness #2**: It seems from the paper that all the questions evaluated in ASQA are ambiguous, and all the questions evaluated in (QA)$^2$ have a false presupposition. This seems to be a somewhat contrived evaluation setup that presumably is unable to test for the ability of the feedback stage to properly identify when an ambiguity or false presupposition is present or absent (the model is rewarded for always saying that the question is ambiguous / contains a false presupposition).
> >
> > **Question:** In the n-shot demonstration setup, how are the n in-context examples sampled? If they are sampled from the same dataset, then the model can effectively "cheat" because it will only ever see in-context examples of ambiguous questions (if the test sample is ambiguous) or questions with false presuppositions (if the test sample has a false presupposition).
>
> **Answer**:
>
> To clarify, we assume that the type of input question is not known in advance. Thus, we randomly sampled the 6 shots from multiple datasets (3 shots from ASQA, and 3 shots from QA$^2$), and used those same few shots for every input question regardless of datasets (ASQA, QA$^2$, and BioASQ).
> These few-shot settings also improved the baseline on ideal and out-of-domain questions (BioASQ).
>
> &nbsp;
>
> > **Weakness #3**: The claim that PreWoMe outperforms GPT4 on ideal questions (L211-212) is also a bit suspect (or at least requires more explanation), given that the method does not substantially differ from baseline question answering in this setting, and the empirical differences are not large.
>
> **Answer**:
>
> As shown in the table above, CoT using "working memory" also improved the baseline on ideal questions, while it is not designed for factoid QA such as BioASQ. CoT promotes LMs to generate intermediate steps of factual knowledge before generating a final answer.
> Similarly, PreWoMe leverages intermediate steps on ideal questions. We conjecture that such intermediate steps elicit world knowledge of LMs, even on biomedical factoid questions in BioASQ.
>
> &nbsp;
>
> > **Question #1**: Qualitatively, how accurate are the generated presuppositions? Is there any correlation between correctly generated presuppositions and answer quality?
>
> **Answer**:
>
> To analyze the accuracy of the generated presuppositions, we randomly sampled 150 questions and manually annotated whether the corresponding 395 generated presuppositions are correct, or not. The accuracy of generated presuppositions was extremely high (Correct 391/total 395, ~99%) demonstrating GPT-4’s capability of generating accurate presuppositions. Since the number of incorrect presuppositions is not statistically significant, we didn't calculate correlations.
>
> Instead, we found the relationship between the generated feedback and the quality of answers. As mentioned in our paper, the feedback acts as a classifier that verbally identifies the type of misleading questions.
> We observed the accuracy on correct and incorrect feedbacks (generated by PreWoMe). For BioASQ and QA$^2$, we report Rouge-L, and for ASQA we report Disambiguation-Rouge.
>
> | | Correct Feed (Ours / Base) | Incorrect Feed (Ours / Base) |
> | ------- | ---------------------- | ------------------------ |
> | BioASQ  | 23.19 / 21.26   | 19.03 / 18.25 |
> | ASQA    | 39.70 / 34.04  | 43.65 / 41.51    |
> | QA$^2$  | 30.45 / 22.38  | 21.93 / 24.02    |
>
> It is noticeable that PreWoMe tends to show higher performances especially when it successfully generates correct feedbacks.
>
> &nbsp;
>
> > **Question #2**: This method seems rather tailored to false presupposition detection, it is unclear to me intuitively whether and why it helps with ambiguity detection. Do the ambiguities usually have to do with one or more of the presuppositions in the question? Can this be quantified in some way, or clarified qualitatively through an example?
>
> **Answer**:
>
> Multiple steps in PreWoMe contribute to detecting the ambiguity in the question.
> We provide some qualitative examples:
>
> &nbsp;
>
> **Example 1:**
> The example below shows feedback/action generation of PreWoMe contributing to detecting ambiguity. PreWoMe effectively detects that the question does not specify which sports World Cup it is referring to, therefore determining that this is an ambiguous question, and providing appropriate action.
> Compared to this, the baseline answers by assuming that the world cup refers to the most famous world cup (FIFA world cup).
>
>
> | Question | Who is hosting the next world cup 2022?|
> | -------- | -------------------------------------------------------------- |
> | PreWoMe  Presuppositions |(1) There is an event called the World Cup (2) The next World Cup is in 2022 (3) The World Cup is hosted by a country (4) There is a clear and single answer to the question.
> |PreWoMe Feedback | The question **does not specify which sport's World Cup is being referred to**. Therefore, the question contains the false assumption that there is a clear and single answer to the question.
> |PreWoMe Action| Your answer should include the host countries for different sports' World Cups in 2022 and provide information about the events and their respective host countries.
> |PreWoMe Prediction| There are several World Cups taking place in 2022 across different sports. Here are some of the major ones: **(1) FIFA World Cup (Soccer)**: The 2022 FIFA World Cup will be hosted by Qatar ………….**(2) ICC Women's Cricket World Cup**: The 2022 ICC Women's Cricket World Cup will be held in New Zealand ………….**(3) FIBA Basketball World Cup (Women)**: The 2022 FIBA Women's Basketball World Cup will be hosted by Australia ………….
> |
> | Baseline | The next FIFA World Cup in 2022 will be hosted by Qatar. ...
> |
>
> &nbsp;
>
> **Example 2:**
> The example below shows generated presuppositions of PreWoMe contributing to detecting ambiguity. PreWoMe generates presuppositions that comprehensively covers the question, by saying that there is a ‘movie’ or ‘song’ called "I’ll be home for Christmas". Compared to this, the baseline just assumes that the question is referring to a song.
>
> | Question | When was i'll be home for christmas released?
> | -------- | -------------------------------------------------------------- |
> | PreWoMe  Presuppositions |(1) There is a **movie or a song** called "I'll Be Home for Christmas".  (2) The movie or song was released at some point in time.  (3) There is a clear and single answer to the question.
> |PreWoMe Feedback | The question contains the false assumption that there is a clear and single answer to the question, as **\"I'll Be Home for Christmas\" could refer to a song or a movie**.
> |PreWoMe Action| Your answer should include the release date of both the song and the movie titled "I'll Be Home for Christmas
> |PreWoMe Prediction| "I'll Be Home for Christmas\" can refer to both a song and a movie. **(1) Song**: "I'll Be Home for Christmas\" is a popular Christmas song written by …………. **(2) Movie**: "I'll Be Home for Christmas" is a 1998 American Christmas family comedy film ………….
> |
> | Baseline | "I'll Be Home for Christmas" is a Christmas song written by …
> |
>
>
> ———————————
>
> **Reference**
>
> [1] Kojima, Takeshi, et al. "Large language models are zero-shot reasoners." Advances in neural information processing systems 35 (2022): 22199-22213.

---

### Meta-Review · Senior_Area_Chairs · 2023-10-05

**Recommendation:** 5

**Metareview:**

The paper suggests a multi-step prompt-based approach for handling open-domain questions, where the questions may contain false-presuppositions or be ambiguous. Specifically, the authors propose to first prompt an LLM for extracting the presuppositions in the question, then prompt it again to decide whether those presuppositions are correct and how to address them in the answer, and finally, given this feedback, generate an answer. The experimental results are sound and the problem is timely and important. Given the above, this paper can be a good fit for the conference.

---

### Decision · Program_Chairs · 2023-10-07

**Decision:**

Accept-Main

**Comment:**

The paper suggests a multi-step prompt-based approach for handling open-domain questions, where the questions may contain false-presuppositions or be ambiguous. Specifically, the authors propose to first prompt an LLM for extracting the presuppositions in the question, then prompt it again to decide whether those presuppositions are correct and how to address them in the answer, and finally, given this feedback, generate an answer. The experimental results are sound and the problem is timely and important. Given the above, this paper can be a good fit for the conference.